# Motion Blur-Free High-Speed Hybrid Image Sensing

**DOI:** 10.3390/s25247496

**Published:** 2025-12-09

**Authors:** Paul K. J. Park, Junseok Kim, Juhyun Ko

**Affiliations:** 1Samsung Electronics, Hwaseong 18448, Gyeonggi-do, Republic of Korea; junseok7.kim@samsung.com (J.K.); juhyun03.ko@samsung.com (J.K.); 2Department of Semiconductor Display, Gachon University, Seongnam 13120, Gyeonggi-do, Republic of Korea

**Keywords:** CMOS image sensor, dynamic vision sensor, hybrid image sensor, motion blur

## Abstract

**Highlights:**

**What are the main findings?**
We introduced a novel homogenous hybrid image sensing technique.The blur image captured by CMOS Image Sensor (CIS) can be compensated effectively.

**What are the implications of the main findings?**
The world’s smallest pixel size of the hybrid image sensor can be achieved.The proposed technique can compensate the motion blur of the CIS image captured in the situation of jogging at a 3 m distance.

**Abstract:**

We propose and demonstrate a novel motion blur-free hybrid image sensing technique. Unlike the previous hybrid image sensors, we developed a homogeneous hybrid image sensing technique including 60 fps CMOS Image Sensor (CIS) and 1440 fps pseudo Dynamic Vision Sensor (DVS) image frames without any performance degradation caused by static bad pixels. To achieve the fast readout, we implemented two one-side ADCs on two photodiodes (PDs) and the pixel output settling time can be reduced significantly by using the column switch control. The high-speed pseudo DVS frame can be obtained by differentiating fast-readout CIS frames, by which, in turn, the world’s smallest pseudo DVS pixel (1.8 μm) can be achieved. In addition, we confirmed that CIS (50 Mp resolution) and DVS (0.78 Mp resolution) data obtained from the hybrid image sensor can be transmitted over the MIPI (4.5 Gb/s four-lane D-PHY) interface without signal loss. The results showed that the motion blur of a 60 fps CIS frame image can be compensated dramatically by using the proposed pseudo DVS frames together with a deblur algorithm. Finally, using the event simulation, we verified that a 1440 fps pseudo DVS frame can compensate the motion blur of the CIS image captured in the situation of jogging at a 3 m distance.

## 1. Introduction

Over the last three decades, the pixel resolution of the CMOS Image Sensor (CIS) has reached to beyond 100 M, which can compete with the human eye [1]. In addition, there is an increasing demand for the development of a motion blur-free high-speed image sensor because the mobile users suffer from image quality degradation due to motion blur, especially in low-illuminance conditions [2]. Recently, it has been proposed that hybrid sensors can mitigate the motion blur by utilizing the event-based high-speed Dynamic Vision Sensor (DVS) [3,4]. However, because these heterogeneous hybrid sensors merged the DVS into a CIS at the pixel level, the image quality of the CIS can be degraded due to DVS pixels (i.e., DVS pixels are considered as deficient pixels in terms of CIS images). Moreover, the DVS pixel size has the fundamental limit of 4 μm due to the large number of required (~30) transistors. To address these problems, we report that a novel homogeneous hybrid sensing technique, having the smallest pseudo DVS pixel size (1.8 μm), can obtain motion blur-free high-speed (~1000 fps) video frames without any performance degradation caused by static bad pixels. In this paper, we propose and demonstrate that the pseudo DVS event data can be extracted by using the frame difference in the fast-readout CIS. The calculation results showed that a 1440 fps pseudo DVS frame (1 Mp) can be transmitted over the interface between sensor and mobile AP together with 60 fps CIS video frames without any signal loss. In addition, we confirmed that the motion blur generated in the CIS video frame could be compensated by using the proposed pseudo DVS event data and deblur algorithm. Section 2 explains the conventional motion-deblur methods, such as two-sensor and heterogeneous hybrid sensor methods. Section 3 describes technical details of the proposed hybrid sensing technique such as operating principle, readout circuit schematic, required data bandwidth, and motion blur compensation. Section 4 discusses the specifications and comparisons with the previous hybrid sensors. In addition, the use case coverage is estimated in Section 4 and conclusions are described in Section 5.

## 2. Conventional Motion-Deblur Methods

### 2.1. Two-Sensor Method

We recently developed a DVS with small pixel sizes (4 μm and 9 μm) [5,6]. The DVS is a unique image sensor inspired by biological visual systems. Each DVS pixel produces a stream of asynchronous events just as the ganglion cells of a biological retina do. By processing the information of local pixels having relative intensity changes instead of entire images at fixed frame rates, the computational requirements can be reduced dramatically. Thanks to the sparsity and binary features of DVS images, the vision task can also be achieved with low computational cost and latency. For example, it has been reported that the motion blur of CIS can be compensated simply by synthesizing with the images obtained from double integral of DVS events [7]. To perform motion blur compensation based on the conventional two-sensor approach, we set up the experiment as shown in Figure 1a. We utilized a moving metal panel stuck on a dead leaves chart as a target subject. The illumination was set to 50 lux and the distance between image sensors and the chart was 1 m. To compensate the motion blur, CIS and DVS cameras were temporally synchronized. In the experiment, we captured the images by adjusting the exposure time to 66.67 ms. Basically, the image quality degradation due to motion blur becomes worse as exposure time is increased. Figure 1b shows that a blurred CIS image is clearly restored by using the deblur technique based on the DVS.

However, during the performance evaluation of motion-deblur based on the conventional two-sensor approach, we found severe artifacts resulting in image quality degradation caused by discrepancies between CISs and DVSs. For example, because a BW (black and white) DVS only outputs the intensity changes, the DVS cannot restore the color information of the CIS accurately, as shown in Figure 2. In particular, in the deblur image, red-color ghosts are observed around the text (Office DEPOT). This is because the pixel resolution, capture time, and optical axis are different between the CIS and DVS. In addition, the DVS has a quantization error, which, in turn, causes ghosts in compensated color intensity due to an imperfect event integral. In addition, the conventional technique could generate various problems such as calibration mismatch, disparity, and occlusions.

### 2.2. Heterogeneous Hybrid Sensor Method

In order to avoid the discrepancy issues in the conventional two-sensor approach, the hybrid sensor has been recently proposed [3,4]. Figure 3 shows the color filter array pattern of the hybrid sensor. The DVS pixels are located in-between the R/B pixels. These pixels adopt white color filters and four PDs are connected and used as one DVS pixel. Because CIS and DVS pixels are separated from each other, this sensor is considered to be a heterogeneous-type hybrid sensor.

The DVS pixel structure is shown in Figure 4. The DVS pixel is composed of a PD, a log trans-impedance amplifier (LOGA), a source follower (SF), a capacitive-feedback amplifier (CFA), two comparators, their following event storages, and in-pixel logic [5]. It results in several tens (~30) of transistors for a pixel, then causes difficulties in reducing the pixel pitch to under 9 μm. This constraint can be resolved by adopting in-pixel Cu-Cu (copper–copper) wafer bonding technology and dividing pixel circuits into top and bottom chips. For example, the circuitry from PD to SF is placed in the top chip, whereas the rest from CFA are placed in the bottom chip. This allows the pixel pitch to be reduced to <5 μm. The DVS pixel size can be reduced further to be around 4 μm by optimizing the allocation of circuitry. However, in principle, the heterogeneous hybrid sensor has a fundamental limit in DVS pixel size of around 4 μm.

## 3. Proposed Hybrid Image Sensing Technique

To mitigate the undesirable artifacts and critical problems of the conventional approaches, we propose and demonstrate a novel homogeneous hybrid sensing technique. Figure 5 shows the operating structure of the proposed hybrid image sensing technique. We developed a 4032 × 3024 CIS with Phase Detection Auto-Focus (PDAF) in all pixels. The size of the PD is 0.9 μm by 1.8 μm. Two PDs compose a single unit pixel. In order to achieve the fast readout, we implemented two ADCs on two PDs. In addition, as for the binning mode, the pixel output settling time can be reduced significantly by using the column switch control for the fast-readout operation, by which, in turn, horizontal random and pattern noises can be decreased as well. As a result, the 1 Mp, 1440 fps CIS images captured by the fast-readout circuit were transmitted to the Image Signal Processing (ISP) chain in the digital logic module. We verified the 1440 fps frame generation by capturing the fabricated sensor chip [8]. The pseudo DVS frame can be generated by differentiating 1440 fps CIS frames [9].Δ*L* = *L_t_* − *L_t_*_−1_ = ln(*Y_t_*/*Y_t_*_−1_)(1)
where *L_t_* and *Y_t_* are log and luma intensity values of the CIS image at time *t*. When the log difference Δ*L* goes beyond the predetermined threshold (i.e., 20%), the pseudo DVS event signal can be generated. Using the numerical simulation based on the frame difference in CIS images [10], we empirically found that 15~30% is the optimum sensitivity range, because it can improve edge sharpness and reduce pixel noises as well. In the experiment, we performed the frame differentiation between 1440 fps CIS frames using the HW emulation board (Simmian board), as shown in Figure 5. Finally, the 60 fps CIS and 1440 fps pseudo DVS frames can be transmitted to the application algorithm module. Thus, the proposed homogeneous hybrid sensing technique can reduce the pseudo DVS pixel size to 1.8 μm, which can break the fundamental limit (~4 μm) of the conventional heterogeneous hybrid sensor.

### 3.1. Fast-Readout Structure

The block diagram of the image sensor is illustrated in Figure 6 [11]. The CIS is exposed to light and the image information is transferred to electric charge by a PD in each pixel. The electric charge of each pixel can be converted to an analog voltage through the readout circuit. Finally, this information is converted to digital code by column ADCs, and then the digital processor improves the image quality. The row and column address decoders are located to generate and enable signals with an address input for sequential data readout.

Figure 7a shows the readout structure of the conventional high-speed CIS [12]. The two-side single slope ADCs had been utilized in the pixel units. However, we utilized 0.9 μm pitch one-side ADCs and optimized the digital processing and floorplan to reduce the power consumption and chip size, as shown in Figure 7b.

Figure 8 shows the equivalent pixel readout circuits. We found that the required current of the proposed readout structure can be four times less than the conventional method. In addition, the time constant (*τ*) of the pixel output of the proposed structure can be described as follows:*τ* = 1/(4√(*K_n_I_c_*) + *R*) × *C*(2)
where *K_n_* is the follower transistor transconductance [13], *I_c_* is the bias current, and *R* and *C* are the resistance and capacitance of the pixel output, respectively. Thus, the time constant of the pixel output derived from the proposed circuit can be decreased by a factor of √2 more than the conventional circuit because the bias current is shared in the pixel readout. Our estimated results showed that the analog power of the readout circuit based on a two-side ADC was 308 mW, whereas a one-side ADC was only about 215 mW in the case of FHD resolution at high-speed (960 fps) operation.

### 3.2. Hybrid Image Generation

Figure 9 shows the operating principles of a hybrid image signal generation. The high-resolution low-speed (60 fps) CIS image can be obtained by merging the 1440 fps images over 24 frames, as shown in Figure 9a. In addition, the low-resolution high-speed (1440 fps) pseudo DVS image can be generated by differentiating the CIS frames as well. In this case, the pseudo DVS frames can be obtained during the time period of the merged CIS frame (i.e., time-shared method). On the other hand, the high-resolution low-speed and low-resolution high-speed images can be captured alternately by switching the pixel readout circuits, as shown in Figure 9b. Thus, the pseudo DVS and CIS frames do not share each other’s capturing time period. Because the time-shared method requires high-resolution and fast-readout operation, the non-time-shared method would be more power-efficient for use by a mobile phone. For the proof of concept, in this paper, we demonstrate the time-shared hybrid sensing technique.

### 3.3. Hybrid Image Transmission

To transmit the hybrid image sensor data over the MIPI interface, we propose a concurrent data transmission technique, as shown in Figure 10. For example, the CIS frame and a series of DVS frames generated in different vertical channels can be merged at the Camera Serial Interface (CSI) block. The CSI-2 block can transmit 10 Gb/s data through a four-lane D-PHY transmitter. The CIS and DVS frame can be interleaved within a transmission frame, and then transmitted over the MIPI interfaces.

In general, the DVS constructs a 2-bit image frame. Thus, each 2-bit per a pixel can be represented as 00 (no event), 01 (ON event), 10 (OFF event), or 11 (not assigned). To packetize DVS event data, we defined two kinds of frame format. Firstly, the raw data transfer mode can send out the 2-bit pixel data, which is similar to the RAW2 format, as shown in Figure 11. Thus, the embedded data of the RAW2 format includes a timestamp and frame number information. This embedded data can be divided into the header and the body. The header size value is fixed at 16 bytes and the data type value is fixed at 1. The frame counter of the header is changed at each frame and increased one by one. In addition, the body parts include the timestamp value of each frame. The same value is repeated to match the total number of bytes of the embedded data.

Secondly, the compressed frame transfer mode can send out the 2-bit pixel data more efficiently than the raw data transfer mode. In this case, DVS event packets can be distinguished by the first 6 bits of each column, as Figure 12 illustrates. The column packet includes the data for the column address, frame number, and frame information. The event group packet represents the data for the group address and group events. Lastly, the frame end packet includes the frame end code for checking data loss. This compressed frame transfer mode can improve the data processing capacity significantly compared to the raw data transfer mode. Previously, a fully synthesized word-serial group address-event representation (G-AER) had been published, which handles massive events in parallel by binding neighboring pixels into a group to acquire data (i.e., pixel events) at high speed, even with high resolution [6]. This G-AER-based compressed method can transmit the sparse event data up to 1.3 Geps at 1 Mp resolution [5].

Figure 13 shows the required bandwidth calculated when the CIS and DVS resolutions are 50 Mp and 0.78 Mp, respectively. In this case, we assumed that the CIS and DVS frame rates were 15 fps and 3000 fps, and the CIS bit depth was 10 bits. It should be noted that the compressed frame transfer mode (G-AER) was considered as DVS frame format [6]. The results show that the required bandwidth was estimated to be 17 Gb/s, which can be transmitted through the MIPI v2.1 (4.5 Gb/s four-lane D-PHY) interface. In the case of using the RAW2 mode for the DVS transmission, the required bandwidth can be calculated to be 12.2 Gb/s. The recent SoC/ISP can support MIPI D-PHY v1.2 interface up to 10 Gb/s. In this case, the maximum required DVS frame rate is 1000 fps (resolution = 1 Mp). However, because the latest image sensor including the MIPI D-PHY v2.1 interface [8] can support 18 Gb/s, the maximum resolution and frame rate of DVS becomes 1 Mp and 3000 fps.

### 3.4. Motion Blur Compensation

Recently, it has been reported that the motion blur of conventional the CIS can be compensated by synthesizing with the images obtained from the DVS [7]. To perform the motion blur compensation, we set up the experiment as shown in Figure 14. We utilized a rotating fan stuck on a Samsung logo as a target subject. We controlled the moving speed by adjusting the motor bias current. The illumination was set to 100–3000 lux and the distance between the image sensor and the fan was 50 cm. When the illumination was less than 50 lux, the pseudo DVS signal could not be observed clearly at 1440 fps. On the other hand, the CIS image became saturated in the case of more than 5000 lux illuminations.

In the experiment, we captured the images by changing exposure times for 1 Mp (CIS and pseudo DVS) images, as shown in Figure 15. We confirmed that the image quality degradation due to motion blur becomes worse as exposure time is increased, whereas the hybrid sensing images are clearly restored by using a deep learning deblur algorithm based on the pseudo DVS. We utilized the Time Lens algorithm (without any modifications for this experiment), which is a CNN framework that combines warping and synthesis-based interpolation approaches to compensate the motion blur [14]. In particular, MTF50 can be defined as the frequency whose spectrum value is decreased to 50% of the low frequency component, which can be used to measure the performance of motion-deblur [15] and the sharpness of YOLO-based object detection [16]. In order to measure the sharpness, we utilized the MTF50 ratio between the ground truth and blurred images. The results showed that the MTF50 ratio was measured to be 0.4 and 0.8 before and after compensation, respectively. It should be noted that the motion-deblur performance based on the proposed pseudo DVS would be same as the heterogeneous hybrid sensor because the pseudo DVS signal is theoretically identical to a real DVS event. In this paper, we demonstrated the 1 Mp resolution CIS motion-deblur and pseudo DVS generation technique. This is because the readout speed of full CIS resolution (12 Mp) is limited to 120 fps, and so 1 Mp resolution was utilized as a concept validation for obtaining a 1440 fps CIS frame. However, for the real chip implementation, as shown in Figure 9, it is required to develop either a whole 12 Mp, 1440 fps fast-readout CIS or a switchable 12 Mp, 60 fps and 1 Mp, 1440 fps CIS.

## 4. Discussion

Figure 16 illustrates the comparisons of specifications. The high-speed CIS of the proposed hybrid sensing technique was implemented in a 65 nm 1P 5M back-illuminated CIS process and a 28 nm 1P 8M standard CMOS process. The top chip includes 1.8 μm pitch 12 M pixels and ADCs and logic circuits are located on the bottom chip. The specifications (pseudo DVS pixel pitch = 1.8 μm) of the proposed sensing technique are superior to the previously published sensors [3,4]. In addition, there is no performance degradation due to bad pixels, unlike the others. Especially, the proposed hybrid sensing technique has the world’s smallest pseudo DVS pixel, which breaks the fundamental limit of 4 μm due to the large number of required (~30) transistors. However, the proposed hybrid sensing is inferior to the heterogeneous hybrid sensor in terms of power consumption and event latency. Specifically, the power consumption of the heterogeneous sensor is less than the proposed hybrid sensing (i.e., 525 mW [4] vs. 845 mW), even though the number of pixels is two times larger (i.e., 2 Mp vs. 1 Mp). This is because the transistors in the DVS circuit are operating in the sub-threshold region [6], while the proposed hybrid sensing technique needs a high-speed readout circuit. Thus, we expect that the power consumption of the proposed technique can be reduced through the optimization of the readout pipeline, the improvement of the high-speed ADC design, and partial row scanning [17]. In addition, the event latency of the proposed sensing technique is larger than the real DVS (i.e., 1 ms vs. 10 μs), due to the CIS frame differentiation. Basically, DVS generates event-driven, frame-free signals asynchronously in response to temporal contrast [18,19,20]. However, there is trade-off relationship between the event latency and power consumption in the proposed hybrid image sensing technique, unlike DVS. For example, to obtain a 10 μs latency in the proposed technique, it is necessary to increase the CIS frame rate up to 100,000 fps. In this case, the power consumption would be beyond several hundred watts.

Figure 17 shows the chip dimension (high-speed CIS only). This chip is fabricated by the SAMSUNG ISOCELL process developed for imaging applications to realize a high-efficiency and low-power photo sensor. The sensor consists of 4032 × 3024 effective pixels that meet with the 1/1.76-inch optical format. The CIS has on-chip 10-bit ADC arrays to digitize the pixel output and on-chip Correlated Double Sampling (CDS) to drastically reduce Fixed Pattern Noise (FPN). It incorporates on-chip camera functions such as defect correction, exposure setting, white balance setting, and image data compression. In addition, the CIS is programmable through a CCI serial interface and includes on-chip One-Time Programmable (OTP) Non-Volatile Memory (NVM). The detailed technical specifications are available online [8].

Lastly, we estimated the use case coverage of the proposed hybrid sensing technique, as shown in Figure 18. Throughout the numerical simulations, we empirically discovered that the motion-deblur tolerance can be described as an MTF50 ratio. For example, when the MTF50 ratio becomes less than 0.7, the image is noticeably blurred. Thus, we defined the deblur score as an MTF50 ratio and the target value was set to 0.7. The motion-deblur depends on the object speed and the distance between the camera and the object. As the object speed increases and the distance decreases, the effect of the motion blur is increased, which, in turn, decreases the deblur score. To compensate this large motion blur, it is required to increase the frame rate of the hybrid sensing technique. Thus, we observed the deblur performance while varying the DVS frame rate and the motion speed of the subject by using the DVS simulator [10] and the event-based double integral technique [7]. Here, a pseudo DVS signal can be numerically generated by using frame differentiation [10] and motion blur can be compensated by using double integral of event signal. Our numerical results show that, because the pseudo DVS frame rate of the proposed hybrid sensing technique is 1440 fps, it can compensate the motion blur of the CIS image captured in the situation of jogging at a distance of almost 3 m. In order to capture the motion blur-free image of a car being driven, our estimation shows that the DVS frame rate should be higher than 5000 fps. In these high-speed use cases (>2000 fps), because the power consumption of the proposed technique would increase beyond several watts, the heterogeneous hybrid sensor would be proper to be used in sports, cars, and military applications. However, toward mobile camera and VR applications, the proposed technique can also be utilized effectively, because there is no image quality degradation due to defective pixels. On top of the motion-deblur, we expect that the high-speed (1440 fps) video frame reconstruction can be achieved by using the proposed hybrid sensing technique and the frame-interpolation algorithm [21].

## 5. Conclusions

We proposed and demonstrated a high-speed (1440 fps) motion blur-free image sensing technique based on hybrid CIS and pseudo DVS frame generation. The proposed hybrid sensing technique has the world’s smallest pseudo DVS pixel compared to its competitors (i.e., 1.8 μm vs. 4.88 μm). In addition, we confirmed that the proposed sensing technique could improve the motion blur performance dramatically without any image quality degradation caused by static bad pixels. However, because the proposed technique utilizes a high-speed readout circuit, the power consumption is larger than the previous heterogenous hybrid sensor (i.e., 845 mW vs. 525 mW). In addition, the event latency of the proposed technique is larger than the real DVS (i.e., 1 ms vs. 10 μs), due to the CIS frame differentiation. Throughout the numerical simulations, we found that the proposed hybrid sensing technique could compensate the motion blur of the CIS image in the situation of jogging at a distance of 3 m (target MTF50 ratio = 0.7). In conclusion, we envision that, as image sensors become even more widespread, for example, with low-light automotive sensing and high-speed action camera, our proposed hybrid sensing technique will also play a critical role in the development of such applications.

## Figures and Tables

**Figure 1 sensors-25-07496-f001:**
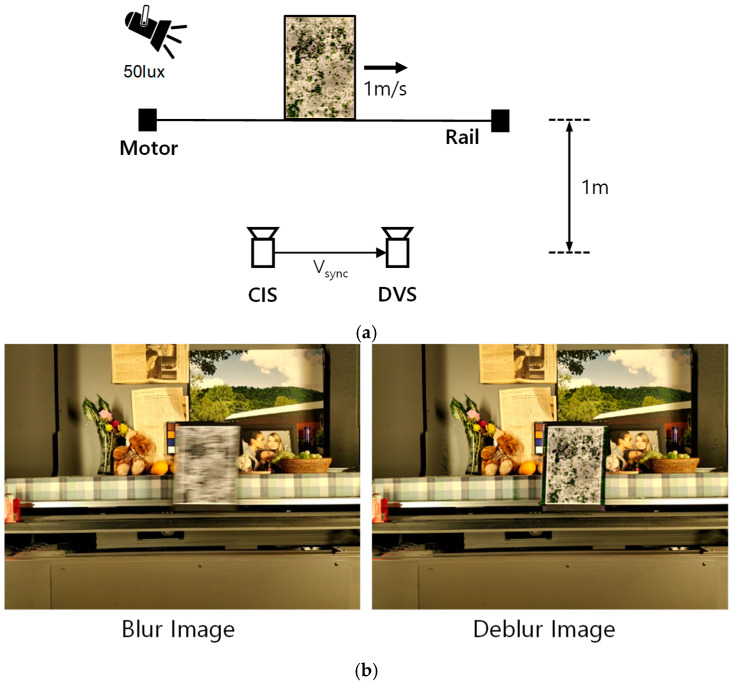
(**a**) Experimental setup and (**b**) deblur results using DVS and event-based double integral technique [7].

**Figure 2 sensors-25-07496-f002:**
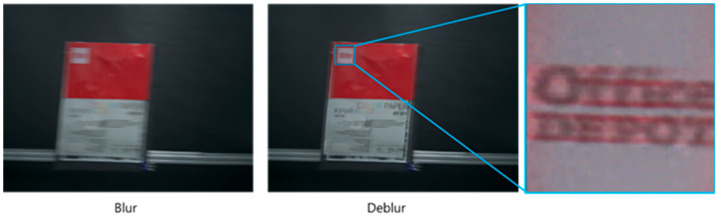
Color information loss of the conventional technique.

**Figure 3 sensors-25-07496-f003:**
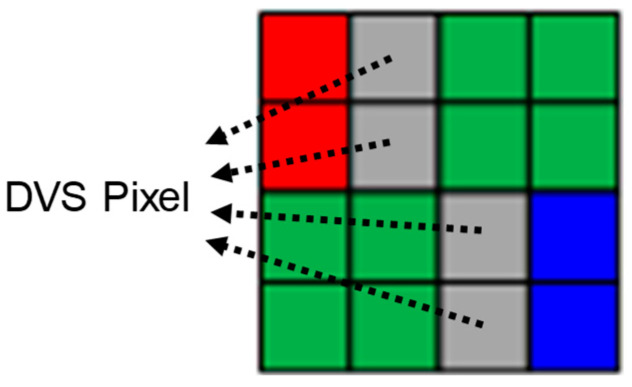
Color filter array pattern of the heterogeneous hybrid sensor [4].

**Figure 4 sensors-25-07496-f004:**
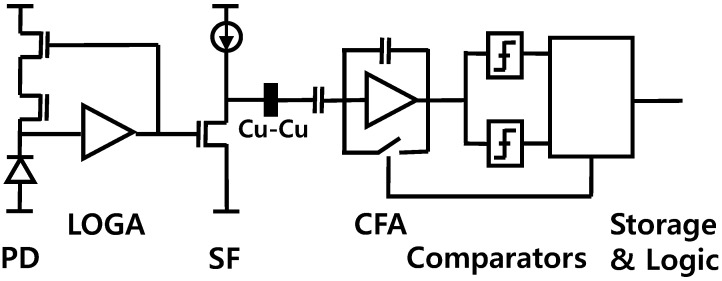
DVS pixel structure.

**Figure 5 sensors-25-07496-f005:**
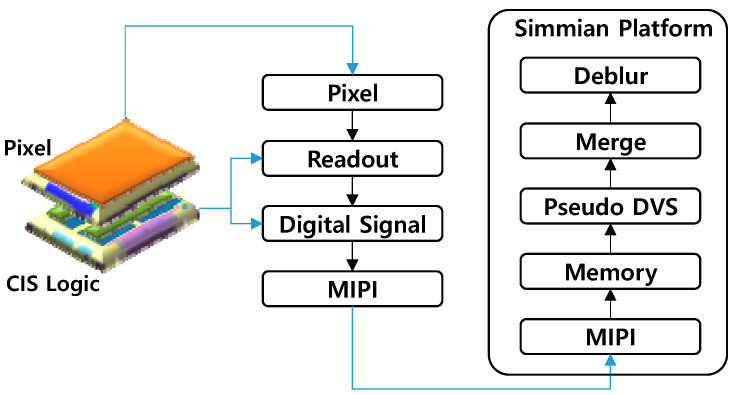
Operating structure of the proposed hybrid sensing technique.

**Figure 6 sensors-25-07496-f006:**
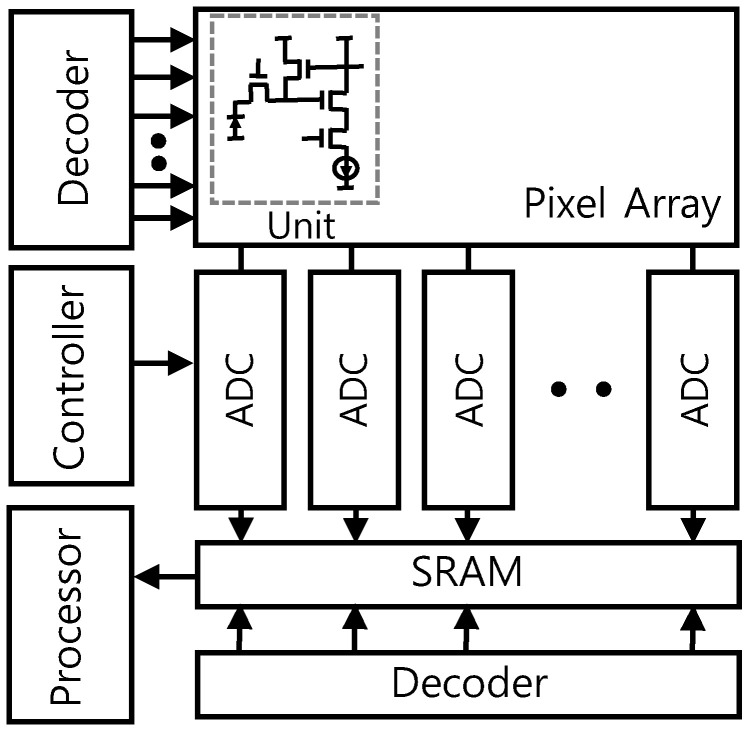
Block diagram of the image sensor.

**Figure 7 sensors-25-07496-f007:**
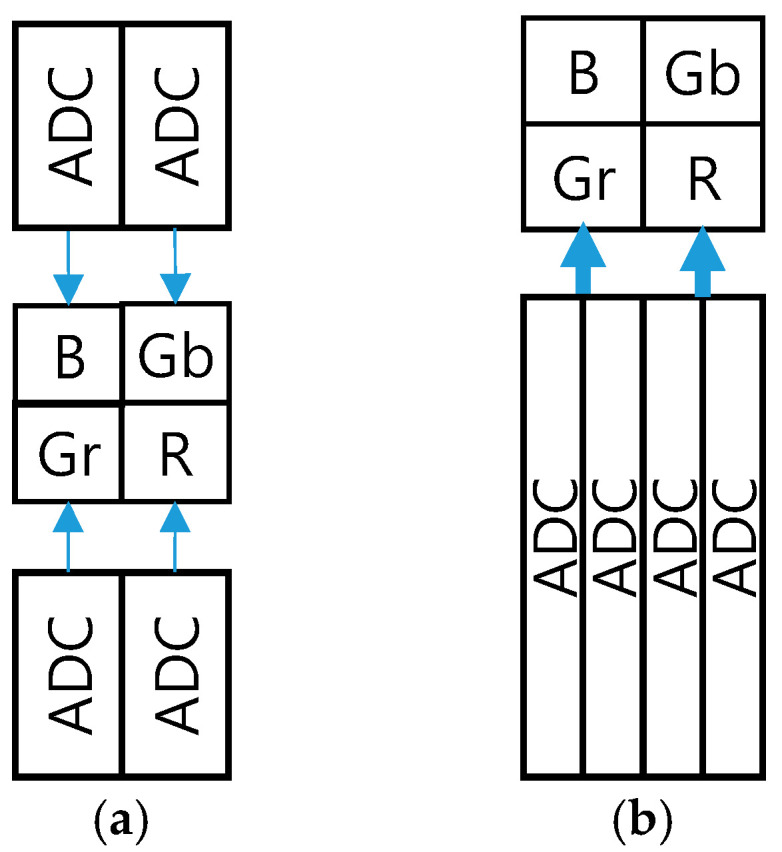
(**a**) Conventional and (**b**) proposed readout structures.

**Figure 8 sensors-25-07496-f008:**
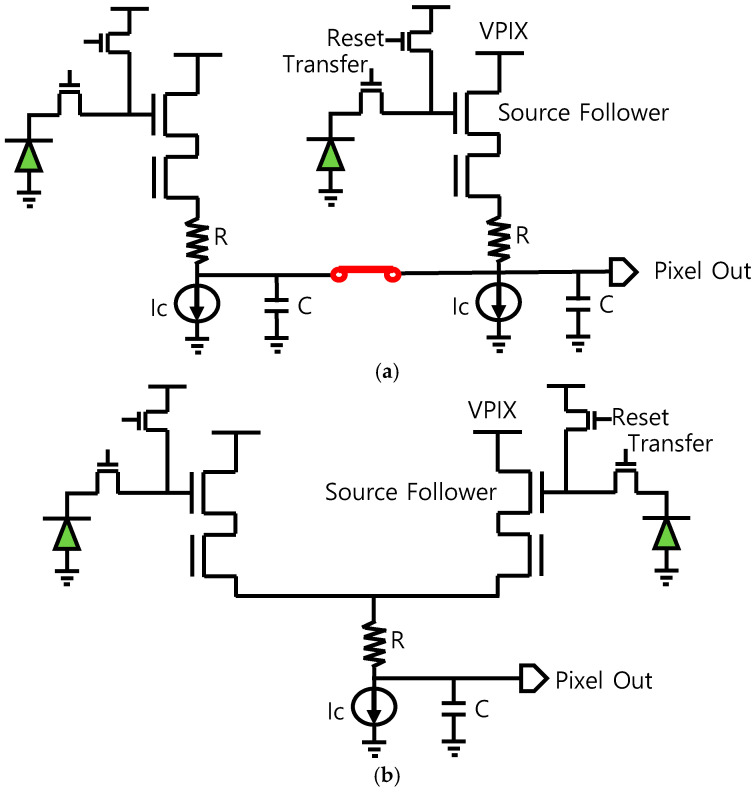
The equivalent pixel readout circuits (**a**) conventional and (**b**) proposed.

**Figure 9 sensors-25-07496-f009:**
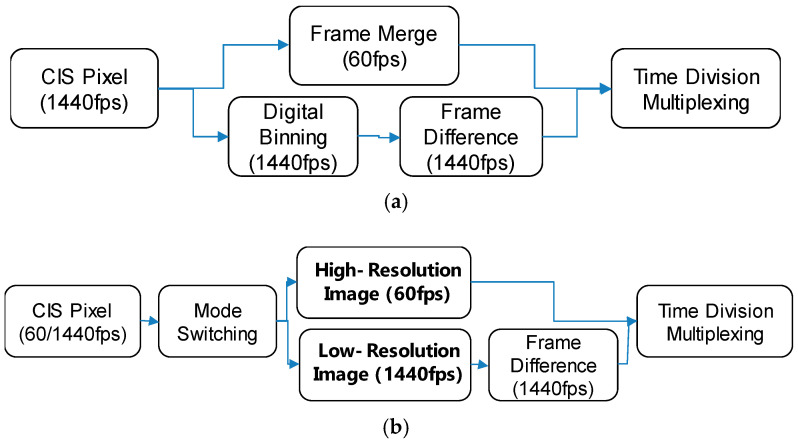
Hybrid image signal generation (**a**) time-shared method and (**b**) non-time-shared method.

**Figure 10 sensors-25-07496-f010:**
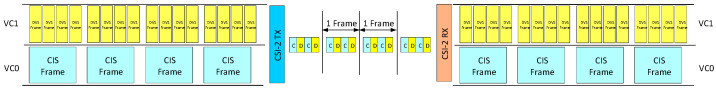
Hybrid image data transmission over the MIPI interface.

**Figure 11 sensors-25-07496-f011:**
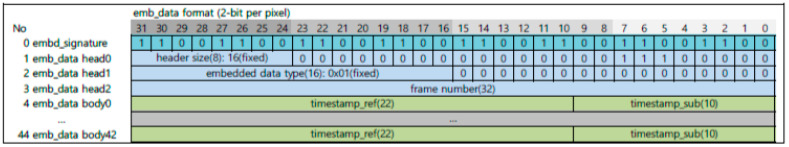
RAW2 embedded header packet.

**Figure 12 sensors-25-07496-f012:**
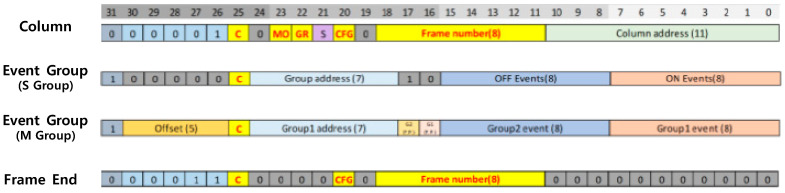
32-bit DVS event packet.

**Figure 13 sensors-25-07496-f013:**
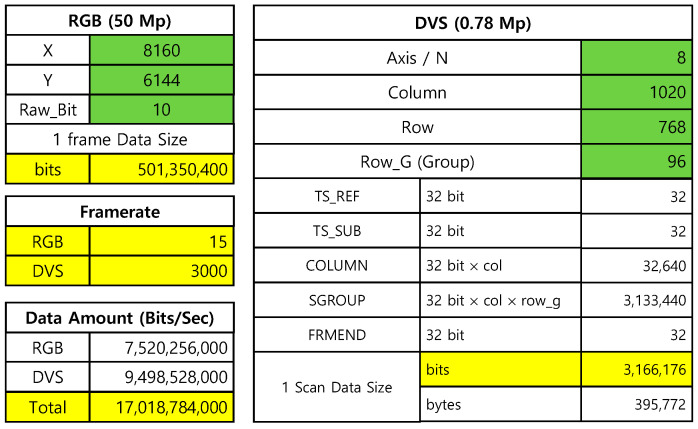
Hybrid image data transmission over the MIPI interface (RGB represents CIS).

**Figure 14 sensors-25-07496-f014:**
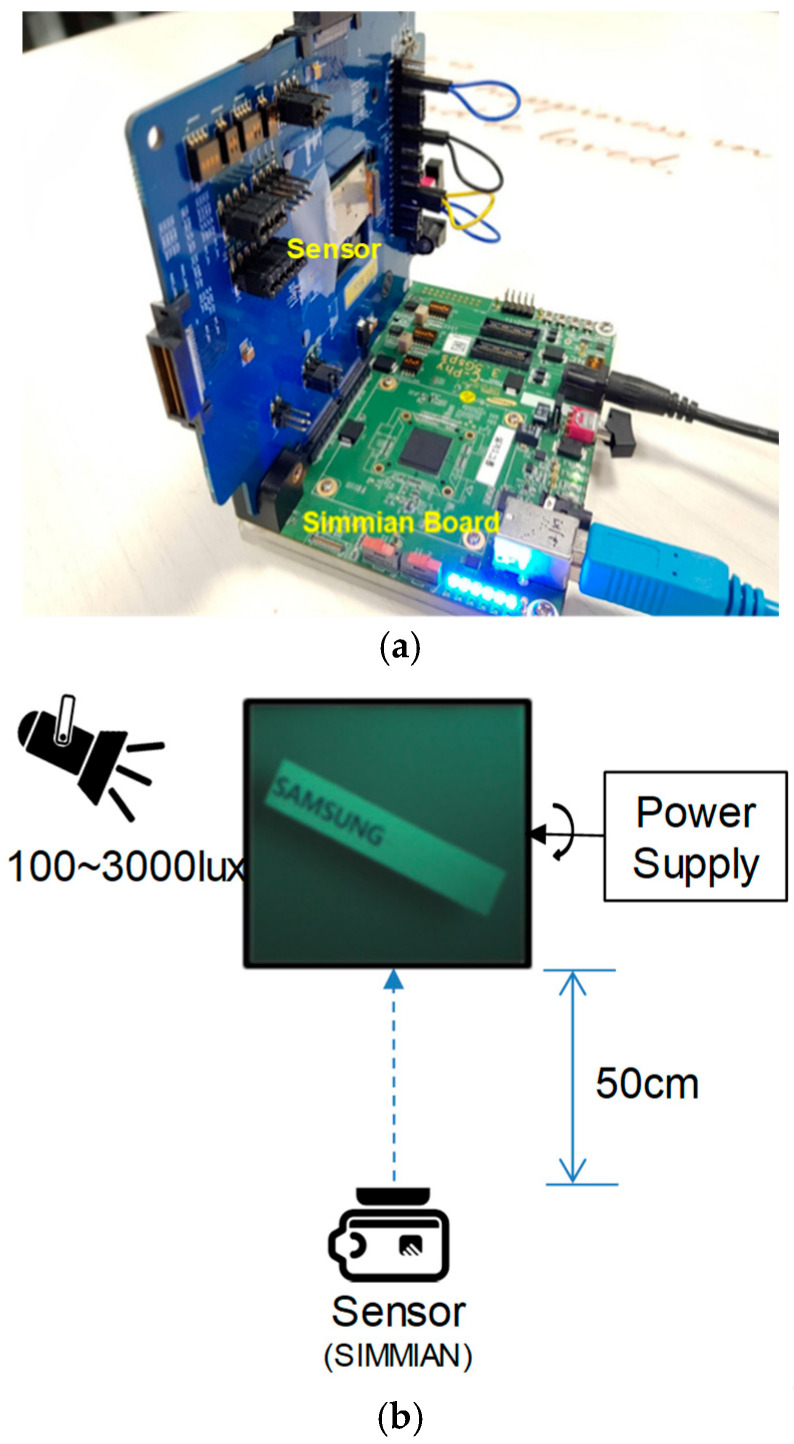
(**a**) Sensor and HW emulation board and (**b**) experimental setup.

**Figure 15 sensors-25-07496-f015:**
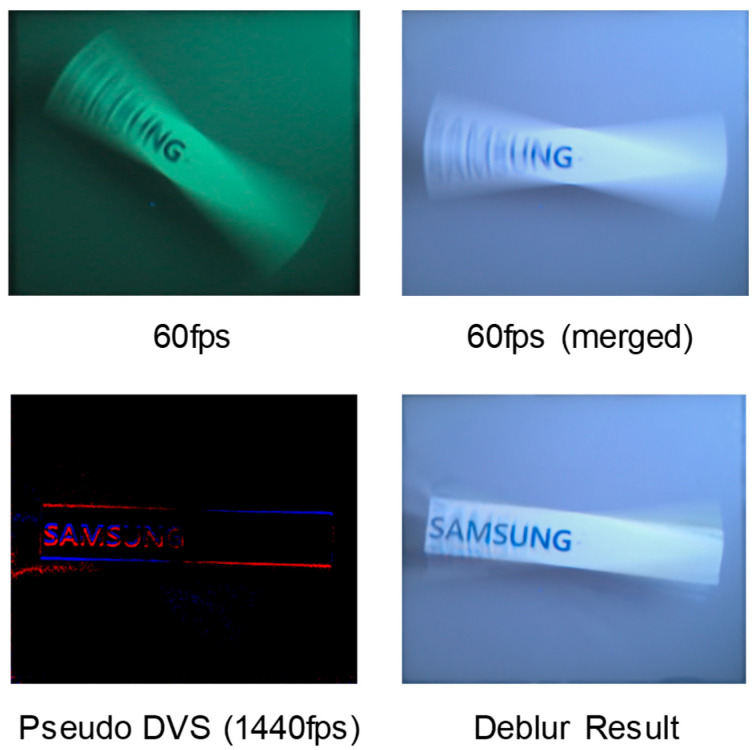
Motion-deblur results using the proposed hybrid image sensing technique (CIS and pseudo DVS resolution: 1 Mp).

**Figure 16 sensors-25-07496-f016:**
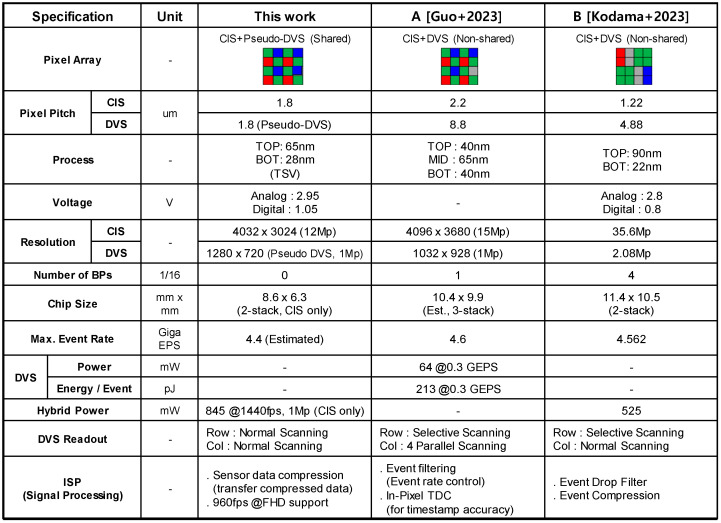
Comparisons of specifications (A [3] and B [4]).

**Figure 17 sensors-25-07496-f017:**
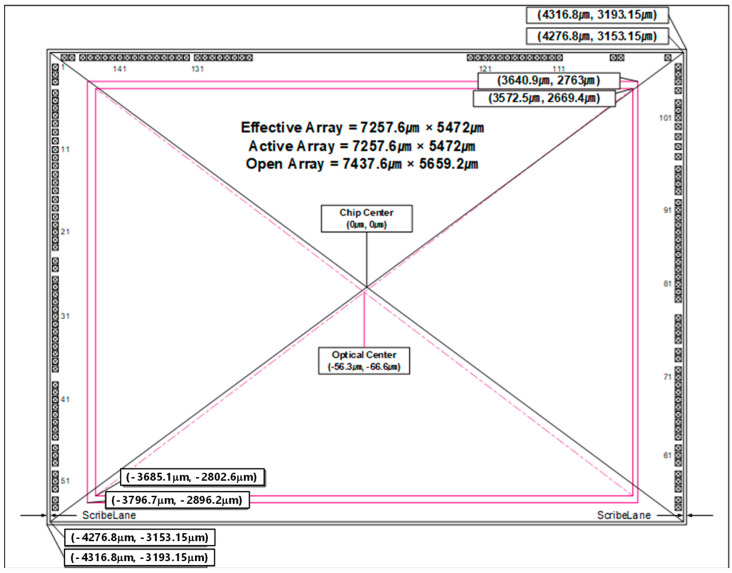
Chip dimension (CIS only, top view).

**Figure 18 sensors-25-07496-f018:**
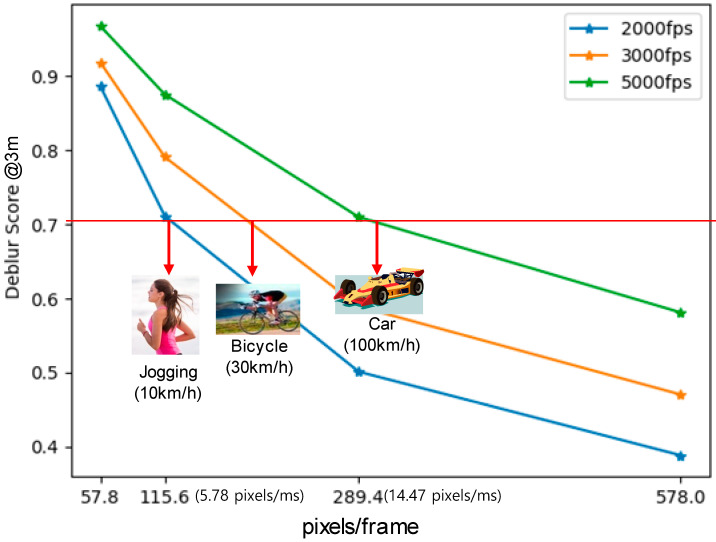
The use case coverage of the motion-deblur simulated with varying DVS frame rates.

## Data Availability

Dataset available on request from the authors.

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
