# Peer review of "Motion Blur-Free High-Speed Hybrid Image Sensing"

_sensors, 2025, doi:10.3390/s25247496_

Round 1

Reviewer 1 Report

Comments and Suggestions for Authors

This manuscript presents a novel homogeneous hybrid image sensing technique that integrates a 60-fps CMOS Image Sensor (CIS) with a 1440-fps pseudo-Dynamic Vision Sensor (DVS) to achieve motion-blur-free imaging. The authors claim several notable contributions, including the world’s smallest pseudo-DVS pixel (1.8 μm), a new fast-readout circuit structure, and the successful transmission of hybrid data over a MIPI interface. The topic is technically relevant and timely, aligning with the growing interest in high-speed, event-based imaging and motion-blur compensation. The work demonstrates a meaningful engineering advance and could be valuable to the imaging sensor community, especially in mobile and low-light applications.

However, several key aspects of the work require further clarification, expansion, or revision before the manuscript can be considered for publication:

  1. Key experimental details are missing or insufficiently described. The “pseudo-DVS frame generation” process via frame differencing needs clearer mathematical formulation and algorithmic description. The deep learning deblur algorithm is mentioned only briefly without specifying network type, training data, or evaluation metrics. Similarly, it is unclear how the 1440-fps frame rate was validated and whether the experiments were conducted on fabricated chips or simulation platforms. Reproducibility currently appears limited.
  2. Most results are qualitative (visual demonstrations of deblurred images). Quantitative comparisons—such as PSNR, SSIM, or motion-blur reduction metrics—against baseline CIS and conventional hybrid sensors are essential to verify performance claims. The authors should also provide power consumption, data bandwidth utilization, and latency measurements under realistic operating conditions. Without such quantitative evidence, the practical advantage of the proposed method remains uncertain.
  3. Several figures (e.g., Figures 5–9) are crowded and contain small or blurry text, making the technical information difficult to interpret. The captions should be rewritten to be self-contained, and schematic diagrams should include more detailed signal flow or timing information. The writing style is sometimes informal and occasionally repetitive; the paper would benefit from careful editing for clarity and conciseness.
  4. For applications of Computer Vision, Some ideas in the articles can be used as reference, including Real-time tracker of chicken for poultry based on attention mechanism-enhanced YOLO-Chicken algorithm, and Pavement distress detection and classification based on YOLO network. Note that if you refer to relevant literature, please add citations appropriately.
  5. The discussion section largely repeats implementation results rather than offering analytical insight. The authors should analyze trade-offs between pseudo-DVS frame rate, pixel size, noise, and bandwidth requirements. Additionally, the “use-case coverage” simulation (Figure 18) is interesting but lacks experimental corroboration—clarify whether this is modeled data and include the assumptions behind it. Connecting these results to broader applications such as automotive or robotics vision could further highlight relevance.

Author Response

Summary

Thank you very much for taking the time to review this manuscript. Please find the detailed responses below and the corresponding revisions/corrections.

Reviewer’s comment. This manuscript presents a novel homogeneous hybrid image sensing technique that integrates a 60-fps CMOS Image Sensor (CIS) with a 1440-fps pseudo-Dynamic Vision Sensor (DVS) to achieve motion-blur-free imaging. The authors claim several notable contributions, including the world’s smallest pseudo-DVS pixel (1.8 μm), a new fast-readout circuit structure, and the successful transmission of hybrid data over a MIPI interface. The topic is technically relevant and timely, aligning with the growing interest in high-speed, event-based imaging and motion-blur compensation. The work demonstrates a meaningful engineering advance and could be valuable to the imaging sensor community, especially in mobile and low-light applications.

However, several key aspects of the work require further clarification, expansion, or revision before the manuscript can be considered for publication:

Comments 1: Key experimental details are missing or insufficiently described. The “pseudo-DVS frame generation” process via frame differencing needs clearer mathematical formulation and algorithmic description. The deep learning deblur algorithm is mentioned only briefly without specifying network type, training data, or evaluation metrics. Similarly, it is unclear how the 1440-fps frame rate was validated and whether the experiments were conducted on fabricated chips or simulation platforms. Reproducibility currently appears limited.

Response 1: Agree. We added mathematical formulation, algorithmic description, and validation of 1440-fps frame rate into the manuscript.

  • (Section 3) The pseudo DVS frame can be generated by differentiating 1440-fps CIS frames [9].

DL = LtLt-1 = ln(Yt/Yt-1)                                                     (1)

where Lt and Yt are log and luma intensity values of CIS image at time t. When the log difference DL becomes to be beyond predetermined threshold (i.e., 20%) then pseudo DVS event signal can be generated.

(reference 9: Hu, Y.; Liu, S.-C.; Delbruck, T. v2e: from video frames to realistic DVS events. In Proceedings of IEEE Conference on Computer Vision and Pattern Recognition Workshops, Nashville, U.S.A., 20 June 2021.)

  • (Section 3.4) We utilized Time Lens algorithm (without any modifications for this experiment) which is a CNN framework, that combines warping and synthesis-based interpolation approaches to compensate the motion blur [13]. In particular, MTF50 can be defined as the frequency whose spectrum value is decreased to 50% of the low frequency component, which can be used to measure the performance of motion deblur [14] and sharpness of YOLO-based object detection [15]. In order to measure the sharpness, we utilized the MTF50 ratio between the ground truth and blurred images. The results showed that the MTF50 ratio was measured to be 0.4 and 0.8 before and after compensation, respectively.

(reference 13: Tulyakov, S.; Gehrig, D.; Georgoulis, S.; Erbach, J.; Gehrig, M.; Li, Y.; Scaramuzza, D. Time lens: event-based video frame interpolation. In Proceedings of IEEE Conference on Computer Vision and Pattern Recognition Workshops, Nashville, U.S.A., 20 June 2021.)

(reference 14: Lin, H.; Mullins, D.; Molloy, D.; Ward, E.; Collins F.; Denny, P.; Glavin, M.; Deegan, B.; Jones, E. Optimizing Camera Exposure Time for Automotive Applications. Sensors 2024, 24, 5135-5163.)

(reference 15: Muller, P.; Braun, A.; Keuper, M.; Examining the impact of optical aberrations to image classification and object detection models. IEEE Transactions on Pattern Analysis and Machine Intelligence 2025, 1-15.)

  • (Section 3) We verified the 1440-fps frame generation by capturing the fabricated sensor chip [8].

(reference 8: ISOCELL 2LD specifications. Available online: https://semiconductor.samsung.com/image-sensor/mobile-image-sensor/isocell-2ld/ (accessed on 21 November 2025))

Comments 2: Most results are qualitative (visual demonstrations of deblurred images). Quantitative comparisons—such as PSNR, SSIM, or motion-blur reduction metrics—against baseline CIS and conventional hybrid sensors are essential to verify performance claims. The authors should also provide power consumption, data bandwidth utilization, and latency measurements under realistic operating conditions. Without such quantitative evidence, the practical advantage of the proposed method remains uncertain.

Response 2: Agree. We added performance matric of motion deblur and technical details into the manuscript.

  • (Section 3.4) In particular, MTF50 can be defined as the frequency whose spectrum value is decreased to 50% of the low frequency component, which can be used to measure the performance of motion deblur [14] and sharpness of YOLO-based object detection [15]. In order to measure the sharpness, we utilized the MTF50 ratio between the ground truth and blurred images. The results showed that the MTF50 ratio was measured to be 0.4 and 0.8 before and after compensation, respectively. It should be noted that the motion deblur performance based on the proposed pseudo DVS would be same as the heterogeneous hybrid sensor because the pseudo DVS signal is theoretically identical to real DVS event.

(reference 14: Lin, H.; Mullins, D.; Molloy, D.; Ward, E.; Collins F.; Denny, P.; Glavin, M.; Deegan, B.; Jones, E. Optimizing Camera Exposure Time for Automotive Applications. Sensors 2024, 24, 5135-5163.)

(reference 15: Muller, P.; Braun, A.; Keuper, M.; Examining the impact of optical aberrations to image classification and object detection models. IEEE Transactions on Pattern Analysis and Machine Intelligence 2025, 1-15.)

  • (Section 4) However, the proposed hybrid sensing is inferior to the heterogeneous hybrid sensor in terms of power consumption and event latency. In detail, the power consumption of heterogeneous sensor is less than the proposed hybrid sensing (i.e., 525 mW [4] vs. 845 mW) even though the number of pixels is two times larger (i.e., 2 Mp vs. 1 Mp). This is because the transistors in DVS circuit are operating in the sub-threshold region [6] while the proposed hybrid sensing technique needs high-speed readout circuit. In addition, the event latency of the proposed sensing technique is larger than the real DVS (i.e., 1 ms vs. 10 ms) due to the CIS frame differentiation.

Comments 3: Several figures (e.g., Figures 5–9) are crowded and contain small or blurry text, making the technical information difficult to interpret. The captions should be rewritten to be self-contained, and schematic diagrams should include more detailed signal flow or timing information. The writing style is sometimes informal and occasionally repetitive; the paper would benefit from careful editing for clarity and conciseness.

Response 3: Agree. Figures 5-9 are simplified and the texts are expressed clearly. In addition, we explained how much the time constant of the readout circuit can be decreased.

  • (Section 3.1) In addition, the time constant (t) of pixel output of the proposed structure can be described as

t = 1/(4Ö(KnIc) + R) ´ C                                                           (2)

   where Kn is the follower transistor transconductance [12], Ic is the bias current, R and C are the resistance and capacitance of pixel output, respectively. Thus, the time constant of pixel output derived from the proposed circuit can be decreased by factor of Ö2 than the conventional circuit because the bias current is shared in the pixel readout.

(reference 12: Salama, K.; Gamal, A. Analysis of active pixel sensor readout circuit. IEEE Transactions on Circuits and Systems 2003, 50, 941-945.)

  • (Section 5) We proposed and demonstrated high-speed (1440 fps) motion-blur-free image sensing technique based on hybrid CIS and pseudo DVS frame generation. The proposed hybrid sensing technique has the world’s smallest pseudo DVS pixel compared to the other competitors (i.e., 1.8 mm vs. 4.88 mm). In addition, we confirmed that the proposed sensing technique could improve the motion blur performance dramatically without any image quality degradation caused by static bad pixels. However, because the proposed technique utilizes frame differentiation, the power consumption is larger than the previous heterogenous hybrid sensor (i.e., 845 mW vs. 525 mW). Throughout the numerical simulations, we found out that the proposed hybrid sensing technique could compensate the motion blur of CIS image in the situation of jogging at a distance of 3m. In conclusion, we envision that as image sensors become even more widespread, for example with low-light automotive sensing and high-speed action camera, our proposed hybrid sensing technique will also play a critical role in the development of such applications.

Comments 4: For applications of Computer Vision, some ideas in the articles can be used as reference, including Real-time tracker of chicken for poultry based on attention mechanism-enhanced YOLO-Chicken algorithm, and Pavement distress detection and classification based on YOLO network. Note that if you refer to relevant literature, please add citations appropriately.

Response 4: Agree. For applications of computer vision, we refer the article YOLO-based object detection in the manuscript.

  • In particular, MTF50 can be defined as the frequency whose spectrum value is decreased to 50% of the low frequency component, which can be used to measure the performance of motion deblur [14] and sharpness of YOLO-based object detection [15].

(reference 14: Lin, H.; Mullins, D.; Molloy, D.; Ward, E.; Collins F.; Denny, P.; Glavin, M.; Deegan, B.; Jones, E. Optimizing Camera Exposure Time for Automotive Applications. Sensors 2024, 24, 5135-5163.)

(reference 15: Muller, P.; Braun, A.; Keuper, M.; Examining the impact of optical aberrations to image classification and object detection models. IEEE Transactions on Pattern Analysis and Machine Intelligence 2025, 1-15.)

Comments 5: The discussion section largely repeats implementation results rather than offering analytical insight. The authors should analyze trade-offs between pseudo-DVS frame rate, pixel size, noise, and bandwidth requirements. Additionally, the “use-case coverage” simulation (Figure 18) is interesting but lacks experimental corroboration—clarify whether this is modeled data and include the assumptions behind it. Connecting these results to broader applications such as automotive or robotics vision could further highlight relevance.

Response 5: Agree. We added performance comparisons, trade-offs, and technical details of use-case coverage into the manuscript.

  • (Section 4) However, the proposed hybrid sensing is inferior to the heterogeneous hybrid sensor in terms of power consumption and event latency. In detail, the power consumption of heterogeneous sensor is less than the proposed hybrid sensing (i.e., 525 mW [4] vs. 845 mW) even though the number of pixels is two times larger (i.e., 2 Mp vs. 1 Mp). This is because the transistors in DVS circuit are operating in the sub-threshold region [6] while the proposed hybrid sensing technique needs high-speed readout circuit. In addition, the event latency of the proposed sensing technique is larger than the real DVS (i.e., 1 ms vs. 10 ms) due to the CIS frame differentiation. Basically, DVS generates event-driven, frame-free signals asynchronously in response to temporal contrast [16-18]. However, there is trade-off relationship between the event latency and power consumption in the proposed hybrid image sensing technique unlike DVS. For example, to obtain a 10-ms latency in the proposed technique, it is necessary to increase CIS frame rate up to 100000 fps. In this case, the power consumption would be beyond several hundred watts.

(reference 16: Brandli, C.; Berner, R.; Yang, M.; Liu, S.-C.; Delbruck, T. A 240 ´ 180 130 dB 3ms latency global shutter spatiotemporal vision sensor. IEEE Journal of Solid-State Circuits 2014, 49, 2333-2341.)

(reference 17: Serrano-Gotarrendona, T.; Linares-Barranco, B. A 128 ´ 128 1.5% contrast sensitivity 0.9% FPN 3ms latency 4mW asynchronous frame-free dynamic vision sensor using transimpedance preamplifiers. IEEE Journal of Solid-State Circuits 2013, 48, 827-838.)

(reference 18: Lenero-Bardallo, J. A.; Serrano-Gotarrendona, T.; Linares-Barranco, B. A 3.6ms latency asynchronous frame-free event driven dynamic-vision-sensor. IEEE Journal of Solid-State Circuits 2011, 46, 1443-1455.)

(Section 4) Lastly, we estimated the use case coverage of the proposed hybrid sensing technique as shown in Figure 18. Throughout numerical simulations, we empirically found out that the motion deblur tolerance can be described as MTF50 ratio. For example, when the MTF50 ratio becomes less than 0.7, the image is noticeably blurred. Thus, we defined the deblur score as MTF50 ratio and the target value was set to 0.7. The motion deblur depends on the object speed and the distance between camera and object. As the object speed increases and the distance decreases, the effect of motion blur is increased, which in turn, the deblur score is decreased. To compensate this large motion blur, it is required to increase the frame rate of hybrid sensing technique. Thus, we observed the deblur performance while varying the DVS frame rate and the motion speed of the subject by using the DVS simulator [19] and the event-based double integral technique [7]. Here, pseudo DVS signal can be numerically generated by using frame differentiation [19] and motion blur can be compensated by using double integral of event signal. Our numerical results show that, because the pseudo DVS frame rate of the proposed hybrid sensing technique is 1440 fps, it can compensate the motion blur of the CIS image captured in the situation of jogging at a distance of almost 3 m. In order to capture the motion-blur-free image of driving car, our estimation shows that the DVS frame rate should be higher than 5000 fps. In this high-speed use cases (>2000 fps), because the power consumption of the proposed technique would increase beyond several watts, the heterogeneous hybrid sensor would be proper to be used in sports, car, and military applications. However, toward mobile camera and VR applications, the proposed technique can be also utilized effectively because there is no image quality degradation due to defective pixels.

Reviewer 2 Report

Comments and Suggestions for Authors

This paper proposes a homogeneous hybrid image sensing technique that utilizes high-speed CMOS fast reading and frame differencing to generate pseudo DVS event signals, and is used to significantly improve motion blur compensation capability. But there are the following issues:

  1. The paper emphasizes the "world's smallest pseudo DVS pixel (1.8 μ m)" and homogeneous hybrid architecture, but lacks quantitative innovation comparison with the latest three-layer stacked DVS-CIS hybrid sensors such as ISSCC/JSCC. It is suggested to supplement more systematic technical benchmarks to strengthen the contribution boundary of the paper.
  2. The frame difference pseudo DVS, time constant reduction, ADC structure optimization, etc. are only provided with block diagrams or conceptual explanations, lacking necessary formulaic descriptions and variable definitions, resulting in insufficient explanation of technical mechanisms.
  3. The analysis of MIPI transmission bandwidth (17 Gb/s) is based on certain assumed parameters and lacks sensitivity analysis for different resolutions, frame rates, RAW2, and compression modes. It is recommended to supplement the feasibility discussion of bandwidth pressure in actual SoC/ISP.
  4. The paper claims that pseudo DVS can simulate real event camera outputs, but does not provide comparative tests of typical DVS indicators such as event sparsity, delay, polarity accuracy, and event noise. It is recommended to add quantitative comparisons with real DVS.
  5. The article uses deep learning to deblurk the model, but does not explain the network structure, training conditions, and whether it is adapted to the pseudo DVS characteristics, which affects reproducibility. Suggest adding necessary algorithm details or providing open source links.
  6. The main experiments focus on rotating fans and small-scale movements, lacking systematic comparisons under conditions of multiple lighting, speeds, backgrounds, and distances. It is recommended to supplement more challenging motion scenes (such as night scenes, handheld shaking, and pedestrian/vehicular traffic scenes).
  7. The deblurring performance should only be presented through visual examples, and quantitative comparisons of indicators such as MTF50, PSNR, SSIM, LPIPS should be included, along with reporting of standard deviation and number of repeated experiments to enhance experimental credibility.
  8. The article only provides partial simulated power consumption (such as 845 mW), but lacks a breakdown of the total system power consumption (ADC, readout link, frame differential module, transmission module, etc.), and does not discuss the heat dissipation and energy efficiency implemented on mobile devices.
  9. The paper proposes two structures, time shared and time non shared, but does not specify engineering considerations such as trade-offs, timing conflicts, and pixel array design complexity. It is suggested to supplement the analysis of architecture design trade-offs.
  10. Some paragraphs are repetitive with the previous content. If the description of the traditional two sensor scheme is lengthy, it can be streamlined and a larger section can be devoted to the technical details and experimental demonstration of the new method.

Author Response

Summary

Thank you very much for taking the time to review this manuscript. Please find the detailed responses below and the corresponding revisions/corrections.

Reviewer’s comment. This paper proposes a homogeneous hybrid image sensing technique that utilizes high-speed CMOS fast reading and frame differencing to generate pseudo DVS event signals, and is used to significantly improve motion blur compensation capability. But there are the following issues:

Comments 1: The paper emphasizes the "world's smallest pseudo DVS pixel (1.8 μ m)" and homogeneous hybrid architecture, but lacks quantitative innovation comparison with the latest three-layer stacked DVS-CIS hybrid sensors such as ISSCC/JSCC. It is suggested to supplement more systematic technical benchmarks to strengthen the contribution boundary of the paper.

Response 1: Agree. We added the comparisons between the proposed technique and heterogeneous hybrid sensor in section 4.

  • (Section 4) However, the proposed hybrid sensing is inferior to the heterogeneous hybrid sensor in terms of power consumption and event latency. In detail, the power consumption of heterogeneous sensor is less than the proposed hybrid sensing (i.e., 525 mW [4] vs. 845 mW) even though the number of pixels is two times larger (i.e., 2 Mp vs. 1 Mp). This is because the transistors in DVS circuit are operating in the sub-threshold region [6] while the proposed hybrid sensing technique needs high-speed readout circuit. In addition, the event latency of the proposed sensing technique is larger than the real DVS (i.e., 1 ms vs. 10 ms) due to the CIS frame differentiation. Basically, DVS generates event-driven, frame-free signals asynchronously in response to temporal contrast [16-18]. However, there is trade-off relationship between the event latency and power consumption in the proposed hybrid image sensing technique unlike DVS. For example, to obtain a 10-ms latency in the proposed technique, it is necessary to increase CIS frame rate up to 100000 fps. In this case, the power consumption would be beyond several hundred watts.

(reference 16: Brandli, C.; Berner, R.; Yang, M.; Liu, S.-C.; Delbruck, T. A 240 ´ 180 130 dB 3ms latency global shutter spatiotemporal vision sensor. IEEE Journal of Solid-State Circuits 2014, 49, 2333-2341.)

(reference 17: Serrano-Gotarrendona, T.; Linares-Barranco, B. A 128 ´ 128 1.5% contrast sensitivity 0.9% FPN 3ms latency 4mW asynchronous frame-free dynamic vision sensor using transimpedance preamplifiers. IEEE Journal of Solid-State Circuits 2013, 48, 827-838.)

(reference 18: Lenero-Bardallo, J. A.; Serrano-Gotarrendona, T.; Linares-Barranco, B. A 3.6ms latency asynchronous frame-free event driven dynamic-vision-sensor. IEEE Journal of Solid-State Circuits 2011, 46, 1443-1455.)

Comments 2: The frame difference pseudo DVS, time constant reduction, ADC structure optimization, etc. are only provided with block diagrams or conceptual explanations, lacking necessary formulaic descriptions and variable definitions, resulting in insufficient explanation of technical mechanisms.

Response 2: Agree. We added the formulaic descriptions of frame differentiation and time constant reduction of ADC structure.

  • (Section 3) The pseudo DVS frame can be generated by differentiating 1440-fps CIS frames [9].

DL = LtLt-1 = ln(Yt/Yt-1)                                                 (1)

where Lt and Yt are log and luma intensity values of CIS image at time t. When the log difference DL becomes to be beyond predetermined threshold (i.e., 20%) then pseudo DVS event signal can be generated.

(reference 9: Hu, Y.; Liu, S.-C.; Delbruck, T. v2e: from video frames to realistic DVS events. In Proceedings of IEEE Conference on Computer Vision and Pattern Recognition Workshops, Nashville, U.S.A., 20 June 2021.)

  • (Section 3.1) In addition, the time constant (t) of pixel output of the proposed structure can be described as

t = 1/(4Ö(KnIc) + R) ´ C                                                           (2)

where Kn is the follower transistor transconductance [12], Ic is the bias current, R and C are the resistance and capacitance of pixel output, respectively. Thus, the time constant of pixel output derived from the proposed circuit can be decreased by factor of Ö2 than the conventional circuit because the bias current is shared in the pixel readout.

(reference 12: Salama, K.; Gamal, A. Analysis of active pixel sensor readout circuit. IEEE Transactions on Circuits and Systems 2003, 50, 941-945.)

Comments 3: The analysis of MIPI transmission bandwidth (17 Gb/s) is based on certain assumed parameters and lacks sensitivity analysis for different resolutions, frame rates, RAW2, and compression modes. It is recommended to supplement the feasibility discussion of bandwidth pressure in actual SoC/ISP.

Response 3: Agree. We added technical details of MIPI transmission bandwidth into the manuscript.

  • (Section 3.3) Figure 13 shows the required bandwidth calculated when the CIS and DVS resolutions are 50 Mp and 0.78 Mp, respectively. In this case, we assumed that CIS and DVS frame rates were 15 fps and 3000 fps, and CIS bit depth was 10 bits. It should be noted that the compressed frame transfer mode (G-AER) was considered as DVS frame format [6]. The results show that the required bandwidth was estimated to be 17 Gb/s, which can be transmitted through the MIPI v2.1 (4.5-Gb/s 4-lane D-PHY) interface. In case of using RAW2 mode for DVS transmission, the required bandwidth can be calculated to be 12.2 Gb/s. The recent SoC/ISP can support MIPI D-PHY v1.2 interface up to 10 Gb/s. In this case, the maximum required DVS frame rate is 1000 fps (resolution = 1 Mp). However, because the latest image sensor including MIPI D-PHY v2.1 interface [8] can support 18 Gb/s then the maximum resolution and frame rate of DVS becomes to be 1 Mp and 3000 fps, respectively.

Comments 4: The paper claims that pseudo DVS can simulate real event camera outputs, but does not provide comparative tests of typical DVS indicators such as event sparsity, delay, polarity accuracy, and event noise. It is recommended to add quantitative comparisons with real DVS.

Response 4: Basically, we think that, to compare the dynamic properties such as event sparsity, delay, polarity accuracy, and event noise between real DVS and pseudo DVS, it should be required to consider same pixel sizes, fill factor, and quantum efficiency. To do this, we need to develop numerical simulators which can simulate the physical properties of DVS and CIS accurately. We will research on this topic further as a next step but, in this paper, we compared power consumption and event latency in section 4 as described in Response 1.

Comments 5: The article uses deep learning to deblur the model, but does not explain the network structure, training conditions, and whether it is adapted to the pseudo DVS characteristics, which affects reproducibility. Suggest adding necessary algorithm details or providing open source links.

Response 5: We utilized Time Lens algorithm (without any modifications for this experiment) which is a CNN framework, that combines warping and synthesis-based interpolation approaches to compensate the motion blur [13].

(reference 13: Tulyakov, S.; Gehrig, D.; Georgoulis, S.; Erbach, J.; Gehrig, M.; Li, Y.; Scaramuzza, D. Time lens: event-based video frame interpolation. In Proceedings of IEEE Conference on Computer Vision and Pattern Recognition Workshops, Nashville, U.S.A., 20 June 2021. (Github open source: https://github.com/uzh-rpg/rpg_timelens))

Comments 6: The main experiments focus on rotating fans and small-scale movements, lacking systematic comparisons under conditions of multiple lighting, speeds, backgrounds, and distances. It is recommended to supplement more challenging motion scenes (such as night scenes, handheld shaking, and pedestrian/vehicular traffic scenes).

Response 6: For the feasibility demonstration of the proposed technique, we performed the experiments of rotating fan. When the illumination becomes to be less than 50 lux, the pseudo DVS signal could not be observed clearly at 1440 fps. On the other hand, the CIS image becomes to be saturated in case of more than 5000-lux illuminations as described in Section 3.4. As for the analysis of challenging motion scenes such as night time, handheld shaking, and pedestrian/vehicle traffic cases, we will research on this topic further as a next step together with the dynamic property comparisons between real and pseudo DVS.

Comments 7: The deblurring performance should only be presented through visual examples, and quantitative comparisons of indicators such as MTF50, PSNR, SSIM, LPIPS should be included, along with reporting of standard deviation and number of repeated experiments to enhance experimental credibility.

Response 7: Agree. We added the quantitative comparisons based on MTF50 into the manuscript. Because we captured the real blur images not using the benchmark data set then we did not get the standard deviation of large data set. However, we confirmed that the same experimental results could be obtained repeatedly in our environment.

  • (Section 3.4) We utilized Time Lens algorithm (without any modifications for this experiment) which is a CNN framework, that combines warping and synthesis-based interpolation approaches to compensate the motion blur [13]. In particular, MTF50 can be defined as the frequency whose spectrum value is decreased to 50% of the low frequency component, which can be used to measure the performance of motion deblur [14] and sharpness of YOLO-based object detection [15]. In order to measure the sharpness, we utilized the MTF50 ratio between the ground truth and blurred images. The results showed that the MTF50 ratio was measured to be 0.4 and 0.8 before and after compensation, respectively. It should be noted that the motion deblur performance based on the proposed pseudo DVS would be same as the heterogeneous hybrid sensor because the pseudo DVS signal is theoretically identical to real DVS event.

(reference 13: Tulyakov, S.; Gehrig, D.; Georgoulis, S.; Erbach, J.; Gehrig, M.; Li, Y.; Scaramuzza, D. Time lens: event-based video frame interpolation. In Proceedings of IEEE Conference on Computer Vision and Pattern Recognition Workshops, Nashville, U.S.A., 20 June 2021.)

(reference 14: Lin, H.; Mullins, D.; Molloy, D.; Ward, E.; Collins F.; Denny, P.; Glavin, M.; Deegan, B.; Jones, E. Optimizing Camera Exposure Time for Automotive Applications. Sensors 2024, 24, 5135-5163.)

(reference 15: Muller, P.; Braun, A.; Keuper, M.; Examining the impact of optical aberrations to image classification and object detection models. IEEE Transactions on Pattern Analysis and Machine Intelligence 2025, 1-15.)

Comments 8: The article only provides partial simulated power consumption (such as 845 mW), but lacks a breakdown of the total system power consumption (ADC, readout link, frame differential module, transmission module, etc.), and does not discuss the heat dissipation and energy efficiency implemented on mobile devices.

Response 8: We added the online access link to check the technical information of the fabricated sensor chip for mobile devices [8]. However, because the frame difference and transmission were performed by using our HW evaluation board as shown in Figure 5, we did not measure the power consumption module by module.

(reference 8: ISOCELL 2LD specifications. Available online: https://semiconductor.samsung.com/image-sensor/mobile-image-sensor/isocell-2ld/ (accessed on 21 November 2025).

Comments 9: The paper proposes two structures, time shared and time non shared, but does not specify engineering considerations such as trade-offs, timing conflicts, and pixel array design complexity. It is suggested to supplement the analysis of architecture design trade-offs.

Response 9: As for the comparisons between time-shared and non-shared structures, we need to study as another research topic. We will research this area as you suggested.

Comments 10: Some paragraphs are repetitive with the previous content. If the description of the traditional two sensor scheme is lengthy, it can be streamlined and a larger section can be devoted to the technical details and experimental demonstration of the new method.

Response 10: Agree. We added the technical details and experimental demonstration of the new method and more discussion as well.

  • (Section 3) The pseudo DVS frame can be generated by differentiating 1440-fps CIS frames [9].

DL = LtLt-1 = ln(Yt/Yt-1)                                                     (1)

where Lt and Yt are log and luma intensity values of CIS image at time t. When the log difference DL becomes to be beyond predetermined threshold (i.e., 20%) then pseudo DVS event signal can be generated.

(reference 9: Hu, Y.; Liu, S.-C.; Delbruck, T. v2e: from video frames to realistic DVS events. In Proceedings of IEEE Conference on Computer Vision and Pattern Recognition Workshops, Nashville, U.S.A., 20 June 2021.)

  • (Section 3.4) We utilized Time Lens algorithm (without any modifications for this experiment) which is a CNN framework, that combines warping and synthesis-based interpolation approaches to compensate the motion blur [13]. In particular, MTF50 can be defined as the frequency whose spectrum value is decreased to 50% of the low frequency component, which can be used to measure the performance of motion deblur [14] and sharpness of YOLO-based object detection [15]. In order to measure the sharpness, we utilized the MTF50 ratio between the ground truth and blurred images. The results showed that the MTF50 ratio was measured to be 0.4 and 0.8 before and after compensation, respectively.

(reference 13: Tulyakov, S.; Gehrig, D.; Georgoulis, S.; Erbach, J.; Gehrig, M.; Li, Y.; Scaramuzza, D. Time lens: event-based video frame interpolation. In Proceedings of IEEE Conference on Computer Vision and Pattern Recognition Workshops, Nashville, U.S.A., 20 June 2021. (Github open source: https://github.com/uzh-rpg/rpg_timelens))

(reference 14: Lin, H.; Mullins, D.; Molloy, D.; Ward, E.; Collins F.; Denny, P.; Glavin, M.; Deegan, B.; Jones, E. Optimizing Camera Expo-sure Time for Automotive Applications. Sensors 2024, 24, 5135-5163.)

(reference 15: Muller, P.; Braun, A.; Keuper, M.; Examining the impact of optical aberrations to image classification and object detection models. IEEE Transactions on Pattern Analysis and Machine Intelligence 2025, 1-15.)

  • (Section 4) Lastly, we estimated the use case coverage of the proposed hybrid sensing technique as shown in Figure 18. Throughout numerical simulations, we empirically found out that the motion deblur tolerance can be described as MTF50 ratio. For example, when the MTF50 ratio becomes less than 0.7, the image is noticeably blurred. Thus, we defined the deblur score as MTF50 ratio and the target value was set to 0.7. The motion deblur depends on the object speed and the distance between camera and object. As the object speed increases and the distance decreases, the effect of motion blur is increased, which in turn, the deblur score is decreased. To compensate this large motion blur, it is required to increase the frame rate of hybrid sensing technique. Thus, we observed the deblur performance while varying the DVS frame rate and the motion speed of the subject by using the DVS simulator [19] and the event-based double integral technique [7]. Here, pseudo DVS signal can be numerically generated by using frame differentiation [19] and motion blur can be compensated by using double integral of event signal. Our numerical results show that, because the pseudo DVS frame rate of the proposed hybrid sensing technique is 1440 fps, it can compensate the motion blur of the CIS image captured in the situation of jogging at a distance of almost 3 m. In order to capture the motion-blur-free image of driving car, our estimation shows that the DVS frame rate should be higher than 5000 fps. In this high-speed use cases (>2000 fps), because the power consumption of the proposed technique would increase beyond several watts, the heterogeneous hybrid sensor would be proper to be used in sports, car, and military applications. However, toward mobile camera and VR applications, the proposed technique can be also utilized effectively because there is no image quality degradation due to defective pixels.

Reviewer 3 Report

Comments and Suggestions for Authors

The authors present a novel homogenous hybrid image sensing technique. This work deals with the compensation of the motion blur of the CMOS Image Sensor (CIS)image captured. The manuscript in general is interesting, however the current state of the manuscript must be improved.  A summary of the rest of the paper should be included at the end of the Introduction section.. This adopts an academic style.

** Do you mean 4µm - 1.8 µm - 9µm.?
** Figure 2. must be described in depth, the authors highlighted (Office Depot) however, this illustration is not detailed in the manuscript. Figure 12 needs to be detailed.
** PDs is used in abstract and defined until line 108. In section 2.2 they used ‘’photodiodes’. Please define in the abstract section and use the acronym in a posterior sentence.
** Cu-Cu (copper-copper)
** Please review equation format in text (4032 x 3024) please use multiplication sign (×). 
**The quality of Figures 7 (b),17l18 are not clear, please review the quality.These Figures must be corrected, due to the quality being too poor.
** Equation of Figure 8 is more appropriate to be explained out of figure. In this figure the equation is not clear.

** Axis X in the fIGURE 18 is pixels/msec or pixels/frame?. Please clear this issue, ms is more appropriate than msec.

**CONCLUSION SECTION
the conclusions are poor, the author must explain in depth with numerical results.. Relevant numerical results could enhance the manuscript.

Author Response

Summary

Thank you very much for taking the time to review this manuscript. Please find the detailed responses below and the corresponding revisions/corrections.

Reviewer’s comment. The authors present a novel homogenous hybrid image sensing technique. This work deals with the compensation of the motion blur of the CMOS Image Sensor (CIS)image captured. The manuscript in general is interesting, however the current state of the manuscript must be improved.  A summary of the rest of the paper should be included at the end of the Introduction section. This adopts an academic style.

  • A summary of the rest of the paper was included at the end of the introduction.

Comments 1: Do you mean 4µm - 1.8 µm - 9µm.?

Response 1: The pixel sizes of reference 5 and 6 are 4 µm and 9 µm, respectively.

Comments 2: Figure 2. must be described in depth, the authors highlighted (Office Depot) however, this illustration is not detailed in the manuscript. Figure 12 needs to be detailed.

Response 2: We explained Figures 2 and 12 in detail.

  • (Section 2.1) Especially, in the deblur image, red-color ghosts are observed around the text (Office DEPOT). This is because the pixel resolution, capture time, and optical axis are different each other between CIS and DVS. In addition, DVS has quantization error, which in turn, causes ghosts in compensated color intensity due to imperfect event integral.
  • (Section 3.3) Previously, it had been published a fully synthesized word-serial group address-event representation (G-AER) which handles massive events in parallel by binding neighboring pixels into a group to acquire data (i.e., pixel events) at high speed even with high resolution [6]. This G-AER-based compressed method can transmit the sparse event data up to 1.3 Geps at 1-Mp resolution [5].

Comments 3: PDs is used in abstract and defined until line 108. In section 2.2 they used ‘’photodiodes’. Please define in the abstract section and use the acronym in a posterior sentence.

Response 3: We defined PD in the abstract and used the acronym in a posterior sentence.

Comments 4: Cu-Cu (copper-copper)

Response 4: We added copper-copper to the sentence.

Comments 5: Please review equation format in text (4032 x 3024) please use multiplication sign (×).

Response 5: We utilized multiplication sign in 4032 ´ 3024.

Comments 6: The quality of Figures 7 (b),17l18 are not clear, please review the quality. These Figures must be corrected, due to the quality being too poor.

Response 6: We changed Figures 7 (b), 17, and 18 with better-quality images.

Comments 7: Equation of Figure 8 is more appropriate to be explained out of figure. In this figure the equation is not clear.

Response 7: We explained the equation of Figure 8 outside of figure.

  • (Section 3.1) Figure 8 shows the equivalent pixel readout circuits. We found out that the required current of the proposed readout structure can be 4 times less than the conventional meth-od. In addition, the time constant (t) of pixel output of the proposed structure can be described as

t = 1/(4Ö(KnIc) + R) ´ C                                                           (2)

where Kn is the follower transistor transconductance [12], Ic is the bias current, R and C are the resistance and capacitance of pixel output, respectively. Thus, the time constant of pixel output derived from the proposed circuit can be decreased by factor of Ö2 than the conventional circuit because the bias current is shared in the pixel readout. Our estimated results showed that the analog power of the readout circuit based on two-sides ADC was 308 mW whereas one-side ADC was only about 215 mW in case of FHD resolution and high-speed (960 fps) operation.

(reference 12: Salama, K.; Gamal, A. Analysis of active pixel sensor readout circuit. IEEE Transactions on Circuits and Systems 2003, 50, 941-945.)

Comments 8: Axis X in the fIGURE 18 is pixels/msec or pixels/frame?. Please clear this issue, ms is more appropriate than msec.

Response 8: It means pixels/ms. msec was replaced by ms.

Comments 9: the conclusions are poor, the author must explain in depth with numerical results. Relevant numerical results could enhance the manuscript.

Response 9: We added numerical results into the conclusions.

  • (Section 5) We proposed and demonstrated high-speed (1440 fps) motion-blur-free image sensing technique based on hybrid CIS and pseudo DVS frame generation. The proposed hybrid sensing technique has the world’s smallest pseudo DVS pixel compared to the other competitors (i.e., 1.8 mm vs. 4.88 mm). In addition, we confirmed that the proposed sensing technique could improve the motion blur performance dramatically without any image quality degradation caused by static bad pixels. However, because the proposed technique utilizes high-speed readout circuit, the power consumption is larger than the previous heterogenous hybrid sensor (i.e., 845 mW vs. 525 mW). In addition, the event latency of the proposed technique is larger than the real DVS (i.e., 1 ms vs. 10 ms) due to the CIS frame differentiation. Throughout the numerical simulations, we found out that the proposed hybrid sensing technique could compensate the motion blur of CIS image in the situation of jogging at a distance of 3 m (target MTF50 ratio = 0.7). In conclusion, we envision that as image sensors become even more widespread, for example with low-light automotive sensing and high-speed action camera, our proposed hybrid sensing technique will also play a critical role in the development of such applications.

Round 2

Reviewer 1 Report

Comments and Suggestions for Authors

My concerns have been addressed, and I appreciate the authors' careful attention to my previous comments and the substantial efforts put forth in revising the manuscript. I am pleased with the updated version, which now provides comprehensive insights and valuable conclusions.

Author Response

Reviewer’s comment. My concerns have been addressed, and I appreciate the authors' careful attention to my previous comments and the substantial efforts put forth in revising the manuscript. I am pleased with the updated version, which now provides comprehensive insights and valuable conclusions.

Response: We would like to express our sincere gratitude for your handling of our manuscript. In revising the manuscript, we carefully considered your comments and concerns. We expanded several sections for clarity, improved the structure and flow of the argument, and refined our analyses and discussion to better highlight the significance of our findings. We are grateful to you for your time and constructive feedback, which greatly contributed to strengthening our work. Thank you again for considering our manuscript for publication.

Reviewer 2 Report

Comments and Suggestions for Authors

The revised draft has completed most of the modifications, but there are still the following issues:

1. It is suggested to further clarify the basis for setting the threshold for pseudo DVS events (refine the rationality of the Δ L threshold by 20%). In 3 In the Proposed Hybrid Image Sensing Technique, the triggering condition for pseudo DVS events, "20% intensity change threshold," is explained after formula (1), but the source or sensitivity analysis of this threshold is missing. Suggest adding an explanation, such as: the threshold comes from empirical experiments, noise statistics results, or belongs to hardware characteristic constraints, to enhance the transparency of the methodology.

2. It is suggested to add a line of explanation about power consumption differences in the Discussion section to make the comparison more complete. In the corresponding paragraph of Figure 16, the author points out that the proposed hybrid sensing technology consumes more power than heterogeneous hybrid sensors (845 mW vs 525 mW). Suggest adding a sentence to explain whether power consumption can be reduced in the future through optimization of the readout pipeline, improvement of ADC design, or partial row scanning.

3. It is recommended to add a brief "limitation statement" to the experimental setup of the Motion Deblur section. In the 3.4 Motion Blur Compensation experiment, 1 Mp resolution CIS and 1440 fps pseudo DVS were used for verification, but the main chip was 12 Mp CIS. Suggest adding an explanation: This is also applicable at high resolution, but limited to chip resources/verification platforms. This study uses 1 Mp resolution as a concept validation. This helps to avoid readers misunderstanding the issue of insufficient experimental scale.

Author Response

Reviewer’s comment. The revised draft has completed most of the modifications, but there are still the following issues:

Comments 1: It is suggested to further clarify the basis for setting the threshold for pseudo DVS events (refine the rationality of the Δ L threshold by 20%). In 3 In the Proposed Hybrid Image Sensing Technique, the triggering condition for pseudo DVS events, "20% intensity change threshold," is explained after formula (1), but the source or sensitivity analysis of this threshold is missing. Suggest adding an explanation, such as: the threshold comes from empirical experiments, noise statistics results, or belongs to hardware characteristic constraints, to enhance the transparency of the methodology.

Response 1: Agree. We added the explanation of the basis for setting the threshold into the manuscript.

  • (Section 3) Using the numerical simulation based on frame difference of CIS images [10], we empirically found that 15~30 % is the optimum sensitivity range because it can improve edge sharpness and reduce pixel noises as well.

Comments 2: It is suggested to add a line of explanation about power consumption differences in the Discussion section to make the comparison more complete. In the corresponding paragraph of Figure 16, the author points out that the proposed hybrid sensing technology consumes more power than heterogeneous hybrid sensors (845 mW vs 525 mW). Suggest adding a sentence to explain whether power consumption can be reduced in the future through optimization of the readout pipeline, improvement of ADC design, or partial row scanning.

Response 2: Agree. We added the sentence suggested by the reviewer to the manuscript.

  • (Section 4) Thus, we expect that the power consumption of the proposed technique can be reduced through optimization of the readout pipeline, improvement of high-speed ADC design, and partial row scanning [17].

(reference 17: Hyun, J.; Kim, H. Low-power CMOS image sensor with multi-column-parallel SAR ADC. Journal of Sensor Science and Technology 2021, 30, 2093-7563.)

Comments 3: It is recommended to add a brief "limitation statement" to the experimental setup of the Motion Deblur section. In the 3.4 Motion Blur Compensation experiment, 1 Mp resolution CIS and 1440 fps pseudo DVS were used for verification, but the main chip was 12 Mp CIS. Suggest adding an explanation: This is also applicable at high resolution, but limited to chip resources/verification platforms. This study uses 1 Mp resolution as a concept validation. This helps to avoid readers misunderstanding the issue of insufficient experimental scale.

Response 3: Agree. We added a brief limitation statement on 1-Mp resolution of pseudo DVS into section 3.4.

  • (Section 3.4) In this paper, we demonstrated 1 Mp-resolution CIS motion deblur and pseudo DVS generation technique. This is because the readout speed of full CIS resolution (12 Mp) is limited to 120 fps then 1-Mp resolution was utilized as a concept validation for obtaining 1440-fps CIS frame. However, for the real chip implementation as shown in Figure 9, it is required to develop either a whole 12-Mp, 1440-fps fast readout CIS or a switchable 12-Mp, 60-fps and 1-Mp, 1440-fps CIS.

Reviewer 3 Report

Comments and Suggestions for Authors

The authors have addressed all comments and suggestions by this reviewer. The current version of this manuscript is better than the submitted version. Many mistakes were detected and corrected by the authors. The manuscript can now be considered suitable for publication.

Author Response

Reviewer’s comment. The authors have addressed all comments and suggestions by this reviewer. The current version of this manuscript is better than the submitted version. Many mistakes were detected and corrected by the authors. The manuscript can now be considered suitable for publication.

Response: We would like to express our sincere gratitude for your handling of our manuscript. We are grateful to you for your time and constructive feedback, which greatly contributed to strengthening our work. Thank you again for considering our manuscript for publication.
